# Impairment of Mesenteric Perfusion as a Marker of Major Bleeding in Trauma Patients

**DOI:** 10.3390/jcm12103571

**Published:** 2023-05-20

**Authors:** Péter Jávor, Tibor Donka, Tamara Horváth, Lilla Sándor, László Török, Andrea Szabó, Petra Hartmann

**Affiliations:** 1Department of Traumatology, University of Szeged, H-6725 Szeged, Hungary; peter.javor.md@gmail.com (P.J.); sandor.lilla.viktoria@med.u-szeged.hu (L.S.); torok.laszlo@med.u-szeged.hu (L.T.); 2Institute of Surgical Research, University of Szeged, H-6724 Szeged, Hungary; horvath.tamara@med.u-szeged.hu (T.H.); szabo.andrea.exp@med.u-szeged.hu (A.S.); 3Department of Sports Medicine, University of Szeged, H-6725 Szeged, Hungary

**Keywords:** hemorrhagic shock, monitoring blood loss, mucosal circulation, superior mesenteric artery perfusion, methane

## Abstract

The majority of potentially preventable mortality in trauma patients is related to bleeding; therefore, early recognition and effective treatment of hemorrhagic shock impose a cardinal challenge for trauma teams worldwide. The reduction in mesenteric perfusion (MP) is among the first compensatory responses to blood loss; however, there is no adequate tool for splanchnic hemodynamic monitoring in emergency patient care. In this narrative review, (i) methods based on flowmetry, CT imaging, video microscopy (VM), measurement of laboratory markers, spectroscopy, and tissue capnometry were critically analyzed with respect to their accessibility, and applicability, sensitivity, and specificity. (ii) Then, we demonstrated that derangement of MP is a promising diagnostic indicator of blood loss. (iii) Finally, we discussed a new diagnostic method for the evaluation of hemorrhage based on exhaled methane (CH_4_) measurement. Conclusions: Monitoring the MP is a feasible option for the evaluation of blood loss. There are a wide range of experimentally used methodologies; however, due to their practical limitations, only a fraction of them could be integrated into routine emergency trauma care. According to our comprehensive review, breath analysis, including exhaled CH_4_ measurement, would provide the possibility for continuous, non-invasive monitoring of blood loss.

## 1. Introduction

Despite the development of trauma care in the past few decades, roughly 25% of post-injury mortality may be potentially preventable by early detection and proper treatment of life-threatening deteriorations. The further development in the management of hemorrhagic shock (HS) is of utmost importance, since HS can be referred as the main cause of potentially preventable trauma mortality [1,2,3].

Regarding the recognition of HS, the challenge is to identify its impending presence in the pre-shock state. To date, the initial hemodynamic assessment of the injured relies on vital signs (VS) such as heart rate, and metabolic markers such as base deficit (BD) and lactate [4,5,6]. However, the specificity of VS and metabolic markers for hypovolemia remained questionable, since several factors such as medication, alcohol intoxication, administration of crystalloids (lactated ringer or saline), or even advanced age can diminish their reliability [7,8,9,10,11]. Furthermore, VS, BD, and lactate are global markers of shock that are maintained at near-normal levels until the compensatory mechanisms of the individual patient become fully exhausted. Consequently, derangements of these indicators during blood loss may remain subtle in the pre-shock state and become apparent when the changes are already non-reversible. In contrast, hemorrhage induces early compensatory mechanisms and temporospatial differences in regional perfusion hallmarked by a redistribution of blood flow from non-vital organs (e.g., the gut and the skin) towards vital vascular beds (i.e., the coronary and cerebral areas) [12,13].

Additionally, the evaluation of the efficacy of treatment is often challenging. Increasing urinary output is a reasonably sensitive marker of improving hemodynamic status; nevertheless, underlying kidney injury, hyperglycemia, or diuretic agents can limit its accuracy [6]. Invasive monitoring methods such as pulmonary artery catheterization offer substantial benefits; however, they are hardly applicable during the initial phase of therapy due to patient positioning and time factor [14,15].

In addition to VS and metabolic markers, hemoglobin (Hb) and hematocrit (Hct) levels are the most frequently used indicators of blood loss due to their several advantages including easy accessibility either with standard laboratory or minimally invasive point of care (POC) testing. However, their diagnostic values in the initial management of trauma patients remains controversial [16]. Initial Hb and Hct levels are influenced by many factors that are not associated with bleeding, such as the patient’s age, gender, weight, and underlying conditions including anemia [17,18]. Furthermore, the on-site Hb values are often lower due to the almost immediate fluid refilling from the interstitium to restore the intravascular volume, early after sustaining trauma. Then, prehospital fluid resuscitation induces further hemodilution and fall in Hct and Hb. Therefore, serial measurements are recommended for the evaluation of trauma-related hemorrhage [18,19], but the results are still controversial [17,20].

Imaging modalities are important adjuncts to the initial hemodynamic assessment in trauma care. Computer tomography (CT) is a reliable method for detecting internal hemorrhage; however, it requires transportation out of the emergency department, resulting in unfavorable time delays. As compared to CT, ultrasound has notable advantages including bedside availability, lack of radiation, reproducibility, and low costs [21]. The focused assessment with sonography in trauma (FAST) and extended FAST (eFAST) protocols can be performed in less than 5 min and display high sensitivity and specificity for hemoperitoneum, hemopericardium, and hemothorax. [22]. Nonetheless, eFAST is hampered by several limitations. Most importantly, the reliability of POC ultrasound depends on the experience of the user and the patient’s body composition. Additionally, visualization of retroperitoneal hemorrhage and differentiation between blood and urine are hardly feasible with ultrasound [21].

Ultimately, no gold standard technique exists for diagnosing and assessing hemorrhage in severe trauma; thus, decision-making is commonly based on a combination of tests, which all have their strengths and limitations. Frequently used tests for the initial hemodynamic assessment of trauma patients are presented in Figure 1, with respect to their accessibility, applicability, sensitivity, and specificity for blood loss scored from 1 to 3 with an arbitrary scaling (Figure 1).

Accessibility refers to the availability of the requirements for implementing the technique, such as machinery, proper instruments, and specially trained personnel. Promptly available vital signs and blood gas parameters obtained the highest accessibility (3). Although urinary catheterization is mostly easy to perform, the hourly diuresis can hardly be determined promptly. Furthermore, catheterization is contraindicated if urethral injury is suspected (e.g., in case of perineal or scrotal hematoma or blood at the meatus) [6]. Due to these limitations, urinary output was judged to carry a moderate level of accessibility (2) in emergency trauma. CT angiography and measurement of pulmonary artery pressure are widely available; nevertheless, they require specific equipment and staff, entailing a medium accessibility (2) for the method. Microcirculatory measurements and monitoring mesenterial blood flow are not routinely utilized techniques and they need a more sophisticated instrumental background. Consequently, their accessibility was ranked the lowest (1).

Applicability is confined to feasibility for continuous monitoring. The parameters and techniques allowing real time monitoring (e.g., pulse oximetry, urinary output, etc.) received the highest applicability (3). The applicability was judged as moderate (2) if monitoring is not a feasible option, but repeated measurements can be performed easily (e.g., blood gas parameters). Although CT angiography can be repeated, executing several CTs during the early phase of patient management is unpractical and potentially dangerous, as it is relatively time consuming and it requires the transportation of the patient to the radiology unit. Therefore, CT angiography obtained the lowest applicability score (1).

Sensitivity refers to the capability used in terms of prompt indication of blood loss and changes in the hemodynamic status of patients. CT angiography, pulmonary artery pressure monitoring, mesenterial blood flow, and microcirculatory changes of gastrointestinal vascular beds are considered as highly sensitive indicators of blood loss [23,24,25]. Additionally, a high sensitivity score (3) was associated with pulse oximetry, as it indubitably reacts to hypoperfusion, although this reaction may be an absent or inaccurate reading. Being more reliable than conventional metabolic markers and vital signs, urinary output, eFAST, Hb, and Hct received medium sensitivity scores (2) [6,21,26,27]. Despite being useful markers in the early assessment of trauma patients, HR, blood pressure, lactate, and BD obtained a sensitivity score of 1 due to the compensatory mechanisms of the body and external influencing factors potentially keeping these values in normal range in the early phases of hemorrhage [6,12].

Specificity refers to the selectivity of the technique for volume depletion and bleeding. Methods directly examining macro- or microvascular systems received the highest specificity value (3). Hb and Hct also obtained a high specificity score (3) for bleeding [16,28,29]. Metabolic markers and cardiovascular vital parameters such as HR and SBP were judged to hold medium specificity for bleeding, since several other factors can influence their values as discussed above. Pulse oximetry is routinely performed to assess blood oxygenation in most emergencies and critical care settings. However, since low perfusion degrades the performance of pulse oximetry, it aims primarily to draw attention to respiratory insufficiencies [30]. For this reason, pulse oximetry obtained the lowest specificity score (1).

Ultimately, it is important to emphasize that the above-described scoring is arbitrary, even though its foundation relies on scientific data. The main goal of this scoring is to illustrate trends and highlight the lack of an easily accessible, highly applicable test with high sensitivity and specificity, calling for further research in the diagnostics of acute blood loss.

Commonly used tests for the initial hemodynamic assessment of trauma patients are presented by highlighting their accessibility, applicability, sensitivity, and specificity for blood loss, based on arbitrary scoring. Mesenterial blood flow refers to the superior mesenteric artery perfusion, while mesenteric mucosal microcirculation concerns specifically microperfusion. Darker colors and higher numbers indicate higher value (easier accessibility, better applicability, higher sensitivity and specificity). Here, accessibility refers to the availability of the requirements for implementing the technique, such as machinery, proper instruments, and specially trained personnel, while applicability is confined to feasibility for continuous monitoring. Sensitivity is capability used in terms of prompt indication of blood loss and changes in the hemodynamic status of patients. Specificity refers to the selectivity of the technique for bleeding; thus, highly specific methods are characterized by reliability for indicating bleeding without being influenced by other factors such as medication, pain, or anxiety. Although vital signs such as heart rate and blood pressure are easily accessible and applicable, they display poor sensitivity and specificity for blood loss. Blood gas parameters, laboratory markers, and imaging modalities provide substantial benefits; however, they do not allow continuous monitoring. Monitoring pulmonary artery pressure and superior mesenteric artery flow blood flow have only the lack of accessibility as a major disadvantage. Making these parameters more accessible in emergency situations by developing prompt and non-invasive techniques to measure them may significantly improve the quality of care.

The reduction in mesenteric perfusion (MP) is among the first compensatory reactions to blood loss, thereby being a potential early clinical indicator of hemorrhage [23,31]. This review offers an insight into the currently available techniques for the evaluation of MP and discusses the possibility of a promising new method that may lead to future quality improvement in emergency trauma care.

## 2. Materials and Methods

### 2.1. Setting the Aim

Some of the authors face the hardships of emergency trauma care day by day. Quality improvement consultations are scheduled on a regular basis in order to designate areas of development in patient care. We agreed that the early recognition of internal hemorrhage is a critical issue that requires improvement in the future. Based on previous studies of our research group, we believed that utilizing the rapid alterations of mesenteric perfusion in circulatory volume depletion has the potential to facilitate this process. For this reason, we decided to provide a comprehensive review of the currently available techniques for the assessment of mesenteric perfusion in the context of emergency trauma.

### 2.2. Data Collection Methods

As a narrative review, data collection was performed from multiple sources, not exclusively from one systematic search. As a first step, the authors listed all methods that were theoretically applicable for the assessment of mesenteric perfusion in the clinical setting. The literature was reviewed by two authors independently, based on a search in MEDLINE (via PubMed) database with the following search terms: “monitoring” AND (“mesenteric perfusion” OR “splanchnic perfusion” OR “intestinal perfusion” OR “mesenteric circulation” OR “splanchnic circulation” OR “intestinal circulation”) AND (“bleeding” OR “haemorrhage” OR “hemorrhage” OR “haemodynamic” OR “hemodynamic”). This search yielded 106 papers which were assessed based on title and abstract. Ultimately, 34 full texts were reviewed. The information obtained from these studies was critically evaluated and extended with data extracted from further papers. Fifty-seven manuscripts were reached via the reference lists of the 34 papers accessed through the above-mentioned MEDLINE search. Forty-nine studies related to the topic were already known by the authors. Nine articles were suggested by other experts the authors briefly consulted with. Twelve studies were accessed through non-systematic use of online search engines. In total, the authors reviewed 161 full-text articles.

## 3. Results

The rapid response of MP is regulated by finely tuned physiological reflexes and neurohumoral processes. As an initial response to a hemorrhage, when the circulating blood volume decreases, the reduction in arterial baroreceptor filling leads to an increased efferent sympathetic activation.

The increased sympathetic output is associated with reflex tachycardia, which, together with the fluid retention via aldosterone and vasopressin, aim to maintain blood pressure. Apart from the cardiac effects, released sympathetic mediators stimulate the α-adrenergic receptors on both the afferent and efferent sides of the microcirculation. Selective vasoconstriction of the afferent arterioles serves to sustain the vascular resistance, while the stimulation of α-adrenergic receptors on postcapillary venules and veins results in autotransfusion by increasing vascular and ultimately the cardiac filling [32].

Arteriolar responses depend on the distribution of the vasoconstrictor α-adrenergic and the vasodilator β2-adrenergic receptor subtypes, which vary within the different tissues. Accordingly, the visceral perfusion is partly sacrificed through the vasoconstrictive response that is mediated by the sympathetic nervous system. However, the abdominal organs are affected unequally by redistribution; for example, intestinal, gastric, and pancreatic blood supplies are more susceptible to the effects of hemorrhage compared to the liver due to the hepatic arterial buffer response [33,34,35]. Intestines are affected by ischemia, particularly adversely and rapidly, due to their unique microanatomy, where the artery and vein within the villi run parallel to each other, which results in low oxygenation in the most luminal areas of the intestine, even under optimal conditions [36,37]. The particular sensitivity of MP to blood loss demonstrated by studies on large animal models, where the superior mesenteric artery (SMA) flow displays a significant drop already at 5% loss of total blood volume; and continues to diminish in parallel with ongoing hemorrhage [23]. Considering the total circulating blood volume as 5 L for an adult, 5% loss means 250 mL of blood, which can hardly be detected with the currently used routine diagnostic tools. This conceptual framework provides the rationale for using biomarkers of the integrity of MP to assess the amount of blood loss in trauma patients.

The almost immediate circulatory redistribution detailed above makes monitoring MP a promising approach in the initial assessment of bleeding trauma patients [23]. Theoretically, there are a wide range of experimentally used methodologies for the evaluation of intestinal macro-and micro perfusion; however, only a fraction of them were integrated into routine emergency trauma care. Methods based on flowmetry, CT imaging, videomicroscopy (VM), measurement of laboratory markers, spectroscopy, tissue capnometry, and breath analysis can all provide valuable information on MP; nevertheless, each technique has its limitations. Figure 2 provides an overview of the currently available methods for assessing MP. In the following, the strengths and limitations of each technique are discussed.

Mesenteric perfusion can be investigated with a variety of techniques. The principles of the methods (diagnostic imaging, flowmetry, VM, laboratory tests, analysis of dissolved and exhaled gases) are shown in white rectangles. The specific techniques or markers are presented in oval text boxes. Based on the capability for the real-time monitoring of MP, techniques can have a static or dynamic nature, which is represented by orange and blue colors. In general, static imaging techniques and laboratory tests reflect the clinical condition of only one moment; thus, they have limited ability in patient monitoring, as bleeding and trauma-related HS are often dynamically progressing conditions. CT = computer tomography, VM = videomicroscopy, OPSI = orthogonal polarization spectral imaging, SDFI = side stream dark field imaging, IDFI = incident dark field imaging, I-FABP = intestinal fatty acid binding protein, IMA = ischemia modified albumin, α-GST = α-glutathione S-transferase, PTRMS = proton transfer reaction mass spectrometry, IFTS = interfacial tensions measurement, PAS = photoacoustic spectroscopy, NIRS = near-infrared spectroscopy, O2C = oxygen-to-see, LDF = laser Doppler flowmetry, CH_4_ = methane.

### 3.1. Diagnostic Imaging

Despite the indubitable value of conventional radiological imaging in trauma care, methods such as extended focused assessment with sonography in trauma (eFAST) and CT cannot be used to monitor the hemodynamic state or MP of trauma patients. It is also important to note that although CT angiography (CTA) is a gold standard for the detection of occlusive mesenteric ischemia, it is hardly capable of providing information on circulatory redistribution during ongoing hemorrhage [37,38,39]. Furthermore, CT is time consuming and requires the transportation of the unstable patient to the radiology unit [39,40]. For these reasons, conventional diagnostic imaging only has limited utility in assessing MP.

### 3.2. Doppler Ultrasound and Laser Doppler Flowmetry

In contrast to conventional diagnostic imaging, Doppler ultrasound (DU) and laser Doppler flowmetry (LDF) are suitable methods for the dynamic visualization of perfusion. Duplex ultrasound combines B-mode and Doppler functions to visualize vessels and their blood flow. Volume flow can be calculated after measuring the cross-sectional area or circumference of the vessel at 90° to the angle of insonation [41]. The LDF technique is based on the Doppler-shift of the reflecting laser beam from moving particles (such as red blood cells (RBCs)) [42]. Both techniques are non-invasive and inexpensive; however, they require superficial targets or artery exposure for precise measurements [43]. Although the assessment of SMA blood flow is, in principle, possible with both DU and LDF; in clinical reality, only newborn patients or intraoperative use can improve their reliability to an acceptable level [44,45,46,47]. As a further limitation of LDF, signals from neighboring large vessels can influence the measurement, resulting in false results [48,49,50].

### 3.3. Videomicroscopic Approaches

Compared to LDF, VM utilizes a different approach by targeting the direct visualization of peripheral microcirculatory networks. There are abundant data supporting the profound disruption of microcirculation in shock, especially in those of septic origin. Therefore, VM is used most commonly as a guide for resuscitation in critical care [51,52,53,54,55,56,57,58,59]. Videomicroscopy allows the bedside assessment of microcirculation by using handheld microscopes to visualize red blood cells in the capillaries of mucosal surfaces [52,54,55]. The first generation of handheld microscopes utilizes orthogonal polarization spectral imaging (OPSI), i.e., polarized light in the wavelength of the spectrum absorption of Hb to detect red blood cells [52]. Subsequently, technological development resulted in the elaboration of side stream dark field imaging (SDFI), a stroboscopic light-emitting diode (LED) ring-based technique allowing better capillary contrast. The second generation of handheld microscopes are SDFI devices (e.g., Microscan (Micro vision Medical B.V., Amsterdam, the Netherlands), CapiScope HVCS (KK Technology, Honiton, UK)) [60,61]. Further improvements were implemented in the third generation of handheld microscopes (incident dark field imaging (IDFI) devices, e.g., CytoCam-IDF (Braedius Medical B.V), which uses a system of 12 high-intensity, short-pulsed LEDs designed to direct the illumination toward the optical axis; and provides greater sensor pixel density [62].

Theoretically, VM is a suitable method for assessing mesenterial microperfusion; however, intestinal mucosa is difficult to access. Nonetheless, if hemodynamic coherence is presumed between the microcirculatory systems of the gut and the sublingual mucosa, sublingual VM is a reasonable approach. Although the association between sublingual and gut microcirculatory networks is supported by evidence [56,57], the reaction of the sublingual area to hemodynamic changes seems to be significantly slower than the response of more distal gastrointestinal regions [23]. Time factor poses an important obstacle to the clinical use of VM in emergency medicine, as it is time-consuming to analyze the records. Furthermore, it may be technically difficult to make the recordings in patients with facial injuries, and results depend on the experience of the examiner.

### 3.4. Laboratory Markers

Contrary to LDF and VM investigating the blood flow directly, laboratory tests aim to detect indirect markers of diminished perfusion. One of the main advantages of laboratory tests is that they provide user-independent, quantitative results with clear cut-off values. Although a variety of MP-specific markers were identified in the past decades, their use in clinical practice is rather uncommon due to several controversies. In the following, the most promising markers including D-lactate, intestinal fatty acid binding protein (I-FABP), ischemia modified albumin (IMA), and α-glutathione S-transferase (α-GST) are discussed.

Mesenteric ischemia is accompanied by the impairment of the gut barrier and possible bacterial translocation [63]. D-Lactate and I-FABP are markers of the integrity and barrier function of the intestinal mucosa [64,65]. D-Lactate is produced mainly by gut bacteria, and its plasma concentration is normally maintained at a concentration of only about 0.01 mm A minor increase in plasma concentration may already indicate intestinal ischemia with enteric bacterial translocation, and a value of 3 mM or higher is known as D-lactic acidosis syndrome [66,67,68]. Intestinal fatty acid binding protein is a cytosolic enzyme present exclusively in enterocytes, and its presence in serum is considered to indicate mucosal injury [67,68]. To date, the evidence supporting the reliability of D-lactate and I-FABP biomarkers in the early diagnosis of HS is scarce; however, promising results with small sample sizes warrant further investigation. A recent study on 26 patients with HS demonstrated significantly elevated I-FABP levels independently from the presence of abdominal injury, compared to a control group of severely injured patients with no HS. The measured I-FABP levels also correlated with clinical parameters for HS such as BD [69]. Nevertheless, it is important to note that despite the potential benefits, I-FABP and D-lactate do not allow continuous monitoring of MP, as they require repeated sampling and laboratory analysis.

Intestinal fatty acid binding protein is not the only cytosolic enzyme with a potential value in assessing intestinal perfusion. α-glutathione S-transferase is an enzyme that is highly active both in the liver and the small intestine mucosa. In addition to its good sensitivity for detecting hepatic ischemia and injury, studies suggest that α-GST may also be useful for diagnosing mesenteric ischemia [70,71,72]. According to a meta-analysis from 2017, the sensitivity and specificity of α-GST for diagnosing acute intestinal ischemia reach 0.68 and 0.84 [73], highlighting moderate potential for providing extra benefit in clinical practice. However, just as other serum biomarkers, α-GST cannot be monitored continuously; moreover, its measurement requires ELISA kits, thus being hardly suitable for emergency situations [74].

Another promising candidate for detecting diminished MP is IMA, an easily accessible marker indicating hypoxic conditions such as pulmonary embolism, acute myocardial infarction, or mesenteric ischemia [74]. In line with the onset of hypoxia, the level of IMA rises rapidly, and then displays a slower, continuous increase for hours [75]. Theoretically, these kinetics makes IMA favorable for the early detection of circulatory redistribution-induced mesenteric ischemia. Nonetheless, the results of animal and clinical studies on IMA were inconsistent. A recent experiment using a hemorrhagic rat model found that IMA and IMA/albumin ratio values followed a similar course to those of lactate; and suggested its use for the early diagnosis of HS under conditions affecting lactate levels [76]. In contrast, other studies found no association between mesenteric ischemia and IMA levels [77]. Based on the currently available literature, IMA may be a useful parameter in the early hemodynamic assessment of trauma patients; however, it is not specific for mesenteric ischemia, and ultimately, it may not provide additional value to current clinical practice.

### 3.5. Measurements of Gas Tensions in Tissues and Exhaled Air

In contrast to laboratory markers, gaseous elements of the human body are suitable subjects for continuous monitoring. Detecting and measuring dissolved or exhaled gases is a well-established, yet rapidly evolving diagnostic field. Within the wide range of techniques, there is a remarkable heterogeneity regarding the target gas, the approach (direct/indirect), and the area of clinical use. In addition to aiding the diagnostics of acute and chronic respiratory illnesses, gastrointestinal ulcers, and lactose intolerance, some gases can provide information also on MP. In the following, the relation between intestinal blood flow and near-infrared spectroscopy (NIRS), micro-lightguide spectrophotometry (O2C), tissue capnometry, and measurement of exhaled methane (CH_4_) is discussed.

#### 3.5.1. Near-Infrared Spectroscopy

Near-infrared spectroscopy (NIRS) is a non-invasive method that is suitable for the continuous, in vivo monitoring of regional tissue oxygenation [78,79]. Similarly, to pulse oximetry, NIRS is based on the modified Beer–Lambert law which relates the attenuation of light to the characteristics of the material through which the light passes [79,80,81]. Since Hb displays different absorption of near-infrared light in response to changes in oxygen levels, changes in tissue oxygenation can be detected with NIRS [82]. Most clinical analyses monitor two different wavelengths utilizing the differential absorption properties of oxygenated and deoxygenated Hb. As a result, an index of oxygenated/deoxygenated Hb can be obtained [83]. In addition to Hb, other biologically important molecules such as albumin [84,85,86], and cholesterol [87,88] can also be investigated using near-infrared light.

In the 1970s, NIRS was originally developed to evaluate cerebral oxygenation; however, it was used for a much wider range of clinical and research purposes in the past two decades [89]. In addition to brain function tests, NIRS is a suitable technique for the quantitative assessment of exercise intolerance in patients suffering from congestive heart failure; furthermore, for evaluating peripheral artery diseases [90], cytochrome c oxidase deficiency [91], metabolic myopathy [92], Friedreich’s ataxia [93], and mitochondrial myopathy [94]. Moreover, apart from the medical field, NIRS has several applications in agriculture as it can provide information on many chemical and physical parameters in crops, fruit, soil, and processed food [95].

Attempts were made for the early recognition of mesenteric ischemia with NIRS, although mainly on animal models and preterm infants at risk of necrotizing enterocolitis [96,97,98,99,100]. The ability of NIRS to detect low mesenteric oxygenation was confirmed; however, the technique is hampered with a number of limitations [101]. It is important to note that Hb and myoglobin have similar optical properties; thus, the extent of the contribution of myoglobin in the measurement sparks controversy. Furthermore, intestines are not located superficial enough in adults to provide a reliable measurement site for a transcutaneous method such as NIRS. Consequently, most studies investigating the potential benefits of NIRS for trauma patients assessed the oxygenation of peripheral musculature instead of intestines [102]. However, skeletal muscles are main stores of myoglobin in the human body [103,104], making the results of these studies even more debatable. Additionally, single-use patient sensors make NIRS monitoring relatively expensive, and comprehensive cost/benefit assessments were not performed for most clinical applications yet [101].

#### 3.5.2. Micro-Lightguide Spectrophotometry (“Oxygen-to-See”/O2C)

Micro-lightguide spectrophotometry is a non-invasive, rapid, and painless method for assessing microvascular circulation [105]. The O2C technique unites backscattering spectroscopy and laser-Doppler flowmetry for measuring oxygen saturation, relative Hb, erythrocyte velocity, and relative blood flow in tissues [105,106,107]. Theoretically, intestinal microcirculation and MP could be assessed with O2C; however, the method shares the main limitation of NIRS as the measurements with the O2C probe are possible only up to a depth of few millimeters [108].

#### 3.5.3. Tissue Capnometry

The measurement of the partial pressure of carbon dioxide (pCO_2_) in tissues is a potentially feasible method for the indirect evaluation of microcirculation [109,110]. Tonometry utilizes the principle that at equilibrium the partial pressure of a diffusible gas such as CO_2_ is equal in the mucosa and in the lumen of a viscus. Thus, gastric tonometry was originally designed to assess splanchnic perfusion in critically ill patients [111], as the stomach is easy to access and is known to be highly sensitive to tissue hypoperfusion [112]. The technique requires the placement of a modified nasogastric tube with a silicone balloon. After luminal pCO_2_ equilibrates the fluid or air in the balloon, CO_2_ is measured via an infra-red CO_2_ analyzer. Thereafter, the discrepancy between gastric pCO_2_ and arterial or end-tidal pCO_2_, the so-called pCO_2_ gap can be calculated. The pCO_2_ gap is suggested to be highly predictive for poor outcome in critically ill patients and patients undergoing major surgery [113,114]. Moreover, a study on six volunteers demonstrated that pCO_2_ gap can indicate hypovolemia before blood pressure, heart rate, lactate, BD, and stroke volume could display any alteration during progressive hemorrhage [115]. Nevertheless, despite being the focus of numerous studies in the 1990s and 2000s, gastric tonometry did not become a routine diagnostic tool in clinical practice [116]. This may be partly a consequence of that gastric tonometry was made commercially available before all of its early methodological issues were resolved and this may have resulted in negative perception [116]. Nonetheless, beyond equivocal reputation, the technique has some practical disadvantages that can hardly be bypassed. Most importantly, the time interval needed for gases to reach equilibrium can be a major hurdle in the emergency setting [117,118]. As well as being time consuming, tube placement can also be an issue of concern. Nasogastric tube insertion can hardly be performed safely on patients with head injuries and potential basilar skull fracture, while orogastric tubes carry additional risks in case of atlanto-occipital dislocation, the most common cervical spine injury related to motor vehicle accidents [119]. Although fiber optic-guided tube insertion may eliminate these risks [120], it would further complicate the method.

The stomach is not the only suitable site of the gastrointestinal tract for tissue capnometry. As gastric microcirculation corresponds to the microcirculation of the sublingual region, measuring pCO_2_ in the sublingual mucosa appears to be a reasonable alternative to gastric tonometry [121,122,123,124]. The difference between pCO_2_ in the sublingual mucosa and arterial pCO_2_ is considered to be predictive of mortality in acute circulatory failure, especially with a cutoff level of 70 mmHg [112,125]. Moreover, the sublingual area is easier to access and free of some limitations of gastric tonometry, such as potential interference of gastric acid [110,126]. The benefits of sublingual capnometry for the management of critically ill or severely injured patients were studied for decades with promising results [12,125,127,128,129,130,131]; however, its clinical use did not become widespread [112]. This may be a consequence of some unelucidated limitations of the method such as the blood-flow-enhancing effect of the device itself through tactile stimuli under the tongue, long equilibration time, and the interference of the CO_2_ production of the oral bacterial flora [132]. Furthermore, prospective, clinical validation studies on large patient populations are also lacking [110,128].

#### 3.5.4. Detection of Exhaled Gases

Breath analysis is a constantly evolving, promising scientific domain being already used routinely for diagnosing pathologies such as lactose intolerance, uremia, or peptic ulcer disease [133,134]. The history of breath testing goes back in time all the way to Hippocrates [135], although its real potential started to unfold with Linus Pauling’s discovery of 250 unique substances present in exhaled breath [136]. The analysis of exhaled gas can be performed on people of all ages and conditions without posing a risk to the patients. Although the potential of breath analysis for the detection and monitoring of mesenteric ischemia is still elusive, attempts were made to test the applicability of the method. A pilot study on rat model aimed to identify volatile markers specific to intestinal ischemia in exhaled breath, and found significantly elevated levels of trimethyldodecatrienol (Z,Z-farnesol-C15H260, 222.37 g/mol MW) during ischemic and reperfusion phases, compared to control measurements [137]. In addition to Z,Z-farnesol, the literature suggests other candidates for extending the list of diagnostic tools for reduced mesenteric blood flow, of which CH_4_ may be the most promising one [23,138]. CH_4_ is an intrinsically non-toxic, combustible gas produced by anaerobic bacterial fermentation [139,140,141]. According to the literature, CH_4_ in the human body originates mainly from methanogenic intestinal microorganisms [66,142]. Due to its physicochemical attributes, CH_4_ can enter freely to the intestinal microcirculation and systemic circulation, and as a gas with low solubility in blood, it becomes rapidly excreted by the lungs [143].

For the measurement of exhaled CH_4_, gas chromatography mass spectrometry is considered as the gold standard technique; however, it does not allow continuous monitoring. Real-time monitoring can be conducted with selected ion flow tube-mass spectrometry, proton transfer reaction mass spectrometry, laser spectrometry, or with photoacoustic spectroscopy (PAS)-based sensors [144,145].

According to the literature, exhaled CH_4_ concentrations correspond to the changes in the blood flow of the SMA [39]. Since SMA perfusion drops significantly already at 5% loss of total blood volume and continues to diminish in parallel with the severity of bleeding [23,138], measuring exhaled CH_4_ levels may offer a new method for the early detection and monitoring of hemorrhage. However, to the best of our knowledge, the validity of this theory was only investigated in animal models so far. A recent study using Vietnamese minipigs (*n* = 6) tested the sensitivity of exhaled CH_4_ for changes in mesenteric macro-and microperfusion during controlled, graded hemorrhage and subsequent fluid resuscitation. Additionally, the performance of this new diagnostic method was compared with sublingual microcirculatory monitoring. The SMAs of the anesthetized, intubated, ventilated animals were accessed from median laparotomy to record blood flow. To provide access to the ileal mucosa for microcirculatory measurements, a 5 cm incision was performed with diathermy 15 cm orally from the ileo–cecal junction. The open mucosal and serosal surfaces were rinsed constantly with saline. Vital signs were monitored continuously during the procedure. CH_4_ concentrations were obtained by attaching a near-infrared laser technique-based PAS apparatus to the exhalation outlet of the ventilator. Hemorrhage was induced and divided into seven phases, followed by gradual fluid resuscitation in five steps, until 80% of the baseline mean arterial pressure value was reached. Each bleeding and resuscitation interval was started with microcirculatory recordings at the ileal mucosal and serosal surfaces and at the sublingual area with IDFI technique (using CytoCam Video Microscope System; Braedius Medical, Huizen, The Netherlands). To quantitatively characterize microcirculation, De Backer score, microvascular flow index, and microvascular heterogeneity index were calculated. The researchers found that diminution in SMA flow and ileal microperfusion were followed rigorously by changes of exhaled CH_4_ levels, and they developed earlier than systemic hemodynamic responses. In contrast, sublingual microcirculation was unable to follow the alterations of MP [23]. These results raise the possibility of a future non-invasive diagnostic and monitoring method in the management of severely injured patients; however, many questions need to be addressed, warranting further research. Since breath analysis does not pose a risk to patients, it is feasible and necessary to conduct human studies. Although swine is considered as the most appropriate animal species for cardiovascular research due to their cardiac anatomy and hemodynamic resemblance to humans [146], it is important to emphasize that the intestinal vascular anatomy and MP of pigs is considerably different [147]. Furthermore, as the above-discussed paper also stated, an important limitation of the method was that some situations did not allow clinicians to obtain baseline CH_4_ values. Consequently, only the alterations of exhaled CH_4_ levels could indicate bleeding, not exact values. Beyond these issues, the influence of thoracal injuries, differences in gut microbiome, and prehospital treatment also need to be elucidated.

## 4. Conclusions

The present review highlighted major difficulties of the initial management of bleeding trauma patients, including the early recognition of HS and the monitoring of therapeutic responses during hemodynamic resuscitation. Ideally, bleeding is identified in the compensatory phase prior to shock; however, the prompt detection of circulatory redistribution often poses a challenge for clinicians. Based on the fact that intestines are affected by hypovolemia particularly adversely and rapidly, we put the diminution of MP, one of the first compensatory reactions to blood loss, into the focus of our study. Theoretically, there are a wide range of experimentally used methodologies for the evaluation of intestinal macro-and microperfusion; however, due to their practical limitations, only a fraction of them were integrated into routine emergency trauma care. The present paper provided an overview on methods based on flowmetry, CT imaging, videomicroscopy (VM), measurement of laboratory markers, spectroscopy, tissue capnometry, and breath analysis, highlighting their strengths and drawbacks.

In the search for a solution to the shortcomings of the currently available methods for assessing MP, we presented a promising new technique, the real-time monitoring of exhaled CH_4_ levels. Studies on animal models demonstrated that exhaled CH_4_ concentrations correspond to the blood flow of the SMA [39], an early indicator of circulatory redistribution. Although animal experiments showed encouraging results, human studies are needed to clarify the relevance and feasibility of this method in clinical practice. A prospective observational study investigating the clinical value of measuring exhaled CH_4_ concentrations in trauma patients is already in progress in our trauma center. Upon completion of the research, the results will be shared with the scientific community through publication in a peer-reviewed journal. In case of a significant association between exhaled CH_4_ and bloodloss, a national multi-center study will be initiated. Additionally, as a near-infrared laser technique-based PAS apparatus can easily be placed in an ambulance car, we also intend to test the method in the prehospital setting. Ultimately, other specialties such as gastroenterology and obstetrics may also benefit from a promptly available non-invasive method indicating circulatory redistribution; thus, the expected benefits clearly justify the thorough investigation of the technique.

## Figures and Tables

**Figure 1 jcm-12-03571-f001:**
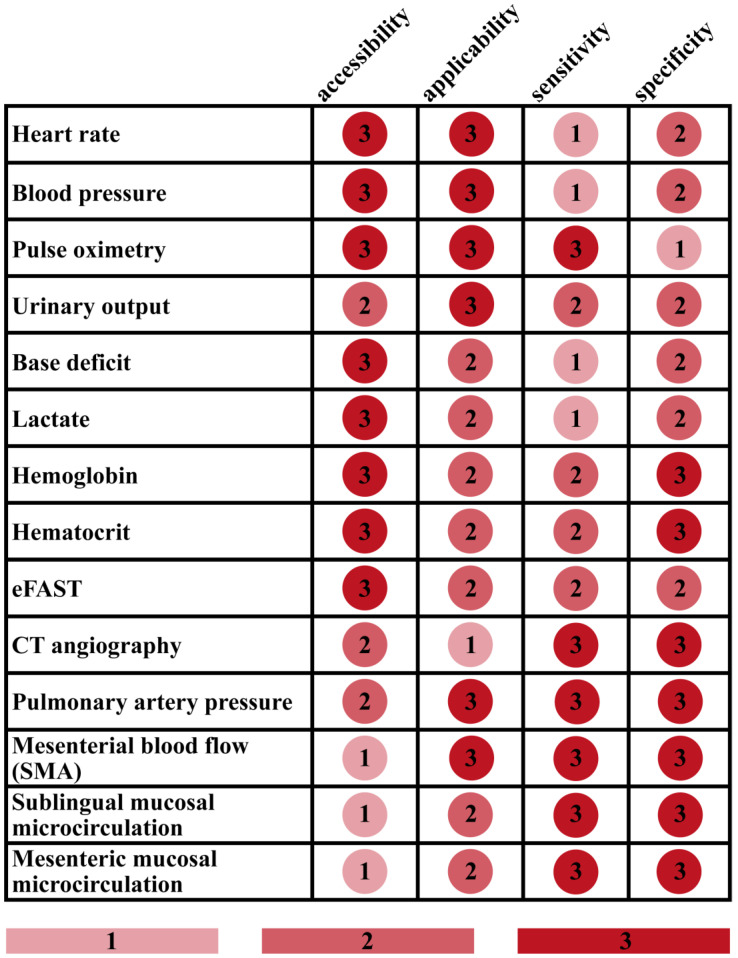
Characterization of frequently used tests for assessing blood loss. eFAST = extended focused assessment with sonography in trauma, CT = computer tomography, SMA = superior mesenteric artery.

**Figure 2 jcm-12-03571-f002:**
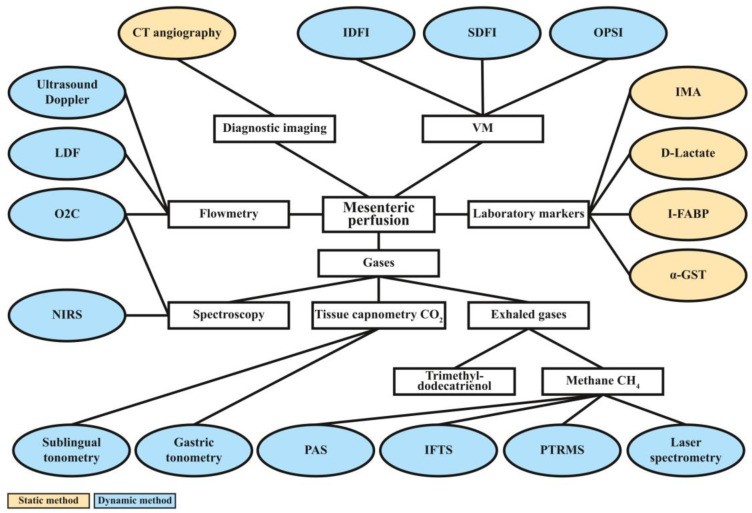
Techniques for the assessment of mesenteric perfusion.

## Data Availability

All the data are available within the article.

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
