# Peer review of "Impairment of Mesenteric Perfusion as a Marker of Major Bleeding in Trauma Patients"

_jcm, 2023, doi:10.3390/jcm12103571_

Round 1
Reviewer 1 Report
Early treatment of major haemorrhage before the onset of shock is associated with improved outcomes. The authors provide strong rationale for measuring impairment of mesenteric microvascular flow, an early compensatory mechanism to preserve blood flow to vital organs, as an indicator of impending shock. The authors review current technologies for assessing mesenteric perfusion, and provide a comprehensive and current overview of clinical, experimental, and animal model data demonstrating the pros and cons of each technology. The authors reasonably conclude that exhaled methane is a candidate marker of mesenteric ischaemia, based on animal data, but requiring further clinical research. This review is well written, appropriately detailed yet accessible to a wider clinical audience. I have no further critique and support publication of this timely review article.
Author Response
Re: Ms. jcm-2386209 - " Impairment of mesenteric perfusion as a marker of major bleeding intrauma patients " by P. Jávor et al.
Reviewer 1#
1.) Early treatment of major haemorrhage before the onset of shock is associated with improved outcomes. The authors provide strong rationale for measuring impairment of mesenteric microvascular flow, an early compensatory mechanism to preserve blood flow to vital organs, as an indicator of impending shock. The authors review current technologies for assessing mesenteric perfusion, and provide a comprehensive and current overview of clinical, experimental, and animal model data demonstrating the pros and cons of each technology. The authors reasonably conclude that exhaled methane is a candidate marker of mesenteric ischaemia, based on animal data, but requiring further clinical research. This review is well written, appropriately detailed yet accessible to a wider clinical audience. I have no further critique and support publication of this timely review article.
We highly appreciate reviewing our manuscript and the positive feedback.
Reviewer 2 Report
The authors present a nice review of early clinical indicators of hemorrhagic shock. Overall the material is relevant, but the organization and presentation needs work.
The manuscript needs to be reorganized. Most of the information presented in the materials and methods section is not relevant to materials and methods. The authors do not describe the methodology they used to perform the review. For this reason, I recommend major revisions.
Some other detailed comments:
Figure 1: Figure needs to include a legend describing colors and numbers or a caption describing them. The central theme of the paper is around reduction of mesenteric perfusion, but it is not clear which of the items in Figure 1 is referring to this phenomenon (“mesenterial blood flow” or “mesenteric mucosal microcirculation” or both? Please clarify.)
Please restrict the Materials and Methods section to the methodology that was used to conduct the review. This information is currently missing from the manuscript.
The animal study of CH4 is missing a reference. See line 402, 454.
Please use consistent formatting of section titles.
Author Response
Re: Ms. jcm-2386209 - " Impairment of mesenteric perfusion as a marker of major bleeding intrauma patients " by P. Jávor et al.
Reviewer 2#
1.) The authors present a nice review of early clinical indicators of hemorrhagic shock. Overall the material is relevant, but the organization and presentation needs work.
The manuscript needs to be reorganized. Most of the information presented in the materials and methods section is not relevant to materials and methods. The authors do not describe the methodology they used to perform the review. For this reason, I recommend major revisions.
Thank you for this observation. The manuscript has been reorganized, and the findings of our literature review have been moved appropriately to the results section. Most importantly, the Materials and Methods section has been extended (page 5, lines 162-185):
“Setting the aim
Some of the authors face hardships of emergency trauma care day by day. Quality improvement consultations are scheduled on a regular basis in order to designate areas of development in patient care. We agreed that the early recognition of internal hemorrhage is a critical issue that requires improvement in the future. Based on previous studies of our research group, we believe that utilizing the rapid alterations of mesenteric perfusion in circulatory volume depletion has the potential to facilitate this process. For this reason, we decided to provide a comprehensive review of the currently available techniques for the assessment of mesenteric perfusion in the context of emergency trauma.
Data collection methods
As a narrative review, data collection was performed from multiple sources, not exclusively from one systematic search. As the first step, the authors have listed all methods that are theoretically applicable for the assessment of mesenteric perfusion in the clinical setting. The literature was reviewed by two authors independently, based on a search in MEDLINE (via PubMed) database with the following search terms: "monitoring" AND ("mesenteric perfusion" OR "splanchnic perfusion" OR "intestinal perfusion" OR "mesenteric circulation" OR "splanchnic circulation" OR "intestinal circulation") AND ("bleeding" OR "haemorrhage" OR "hemorrhage" OR "haemodynamic" OR "hemodynamic"). This search yielded 106 papers which were assessed based on title and abstract. Ultimately, 34 full texts were reviewed. The information obtained from these studies was critically evaluated and extended by expert opinions and data from papers reviewed through other methods (e.g. papers that were already well-known by the authors, studies that were cited in the papers of the MEDLINE search mentioned above, etc.).
2.) Figure 1: Figure needs to include a legend describing colors and numbers or a caption describing them.
Thank you for this remark. Colors and numbers are briefly described in the figure legend and we have extended the manuscript with a detailed description of our arbitrary scoring system (pages 2-3, lines 78-132; page 4, lines 135-154).
3.) The central theme of the paper is around reduction of mesenteric perfusion, but it is not clear which of the items in Figure 1 is referring to this phenomenon (“mesenterial blood flow” or “mesenteric mucosal microcirculation” or both? Please clarify.)
Thank you for this notification. Both items are associated with mesenteric perfusion: “mesenterial blood flow” refers to macroperfusion - the blood flow of the superior mesenteric artery; while “mesenteric mucosal microcirculation” refers specifically to microperfusion - the microcirculatory system of the mesenteric mucosa. This has also been clarified on Figure 1 and in the legend of the figure (figure 1, page 4, lines 135-154):
“Mesenterial blood flow (SMA)”
“Mesenterial blood flow refers to the superior mesenteric artery perfusion, while mesenteric mucosal microcirculation concerns specifically microperfusion.”
4.) Please restrict the Materials and Methods section to the methodology that was used to conduct the review. This information is currently missing from the manuscript.
Thank you for this remark. Please see our answer above, under comment number 1.)
5.) The animal study of CH4 is missing a reference. See line 402, 454.
Thank you for this comment. The reference has been placed accordingly. (page 13, lines 492, 522)
6.) Please use consistent formatting of section titles.
Thank you for this suggestion. The manuscript has been reformatted accordingly.
Reviewer 3 Report
Dear author,
thanks for the nice review, especially of the new monitoring options. However, I think that these should be substantiated more with numbers, especially sensitivity, specificity, average time to availability, cost-benefit, availability in trauma centers, learning curve of the users, so that these methods are also increasingly used in the clinic at some point.
Especially in Figure 1, real numbers would be useful here instead of a scale of 1-3, which is not clearly defined.
Author Response
Re: Ms. jcm-2386209 - " Impairment of mesenteric perfusion as a marker of major bleeding intrauma patients " by P. Jávor et al.
Reviewer 3#
1.) Dear author,
thanks for the nice review, especially of the new monitoring options. However, I think that these should be substantiated more with numbers, especially sensitivity, specificity, average time to availability, cost-benefit, availability in trauma centers, learning curve of the users, so that these methods are also increasingly used in the clinic at some point. Especially in Figure 1, real numbers would be useful here instead of a scale of 1-3, which is not clearly defined.
Thank you for reviewing our manuscript. Your valuable remark made us realize that our arbitrary ranking of diagnostic markers and methods has not been clearly defined. We have striven to correct this shortcoming and provided detailed explanation for our scoring. A minor correction was also made in Figure 1 itself: the sensitivity score of hematocrit was corrected from 1 to 2, since we noticed that it had incidentally received a lower score due to an editing mistake (the row was shifted during the editing the of the figure).
(The above-mentioned changes can be found in: Figure 1 and pages 2-3, lines 78-132):
“Frequently used tests for the initial hemodynamic assessment of trauma patients are presented in Figure 1, with respect to their accessibility, applicability, sensitivity, and specificity for blood loss scored from 1 to 3 with an arbitrary scaling (Fig. 1).
Accessibility refers to the availability of the requirements for implementing the technique, such as machinery, proper instruments, and specially trained personnel. Promptly available vital signs and blood gas parameters got the highest accessibility (3). Although urinary catheterization is mostly easy to perform, the hourly diuresis can hardly be determined promptly. Furthermore, catheterization is contraindicated if urethral injury is suspected (e.g. in case of perineal or scrotal hematoma or blood at the meatus) [6]. Due to these limitations, urinary output has been judged to carry a moderate level of accessibility (2) in emergency trauma. CT angiography and measurement of pulmonary artery pressure are widely available; nevertheless, they require specific equipment and staff, entailing a medium accessibility (2) for the method. Microcirculatory measurements and monitoring mesenterial blood flow are not routinely utilized techniques and they need a more sophisticated instrumental background. Consequently, their accessibility was ranked the lowest (1).
Applicability is confined to feasibility for continuous monitoring. The parameters and techniques allowing real time monitoring (e.g. pulse oximetry, urinary output, etc.) received the highest applicability (3). The applicability was judged as moderate (2) if monitoring is not a feasible option, but repeated measurements can be performed easily (e.g. blood gas parameters). Although CT angiography can be repeated, executing several CTs during the early phase of patient management is unpractical and potentially dangerous as it is relatively time consuming and it requires the transportation of the patient to the radiology unit. Therefore, CT angiography got the lowest applicability score (1).
Sensitivity refers to the capability used in terms of prompt indication of blood loss and changes in the hemodynamic status of patients. CT angiography, pulmonary artery pressure monitoring, mesenterial blood flow, and microcirculatory changes of gastrointestinal vascular beds are considered as highly sensitive indicators of blood loss [23-25]. Additionally, a high sensitivity score (3) was associated with pulse oximetry, as it indubitably reacts to hypoperfusion, although this reaction may be an absent or inaccurate reading. Being more reliable than conventional metabolic markers and vital signs, urinary output, eFAST, Hb and Hct received medium sensitivity score (2) [6,21,26,27]. Despite being useful markers in the early assessment of trauma patients, HR, blood pressure, lactate and BD obtained a sensitivity score of 1 due to the compensatory mechanisms of the body and external influencing factors potentially keeping these values in normal range in the early phases of hemorrhage [6,12].
Specificity refers to the selectivity of the technique for volume depletion and bleeding. Methods directly examining macro- or microvascular systems received the highest specificity value (3). Hb and Hct also obtained a high specificity score (3) for bleeding [16,28,29]. Metabolic markers and cardiovascular vital parameters such as HR and SBP were judged to hold medium specificity for bleeding, since several other factors can influence their values as discussed above. Pulse oximetry is routinely performed to assess blood oxygenation in most emergencies and critical care settings. However, since low perfusion degrades the performance of pulse oximetry, it aims primarily to draw attention to respiratory insufficiencies [30]. For this reason, pulse oximetry obtained the lowest specificity score (1).
Ultimately, it is important to emphasize that the above-described scoring is arbitrary, even though its foundation relies on scientific data. The main goal of this scoring is to illustrate trends and highlight the lack of an easily accessible, highly applicable test with high sensitivity and specificity, calling for further research in the diagnostics of acute blood loss.”
The references listed below were added to the reference list to support the scientific data on basic research and clinical measurement methodologies:
- Jávor, P.; Hanák, L.; Hegyi, P.; Csonka, E.; Butt, E.; Horváth, T.; Góg, I.; Lukacs, A.; Soós, A.; Rumbus, Z.; et al. Predictive Value of Tachycardia for Mortality in Trauma-Related Haemorrhagic Shock: A Systematic Review and Meta-Regression. BMJ Open 2022, 12, doi:10.1136/BMJOPEN-2021-059271.
- Dubin, A.; Pozo, M.O.; Ferrara, G.; Murias, G.; Martins, E.; Canullán, C.; Canales, H.S.; Kanoore Edul, V.S.; Estenssoro, E.; Ince, C. Systemic and Microcirculatory Responses to Progressive Hemorrhage. Intensive Care Med 2009, 35, 556–564, doi:10.1007/S00134-008-1385-0.
- Hadian, M.; Pinsky, M.R. Evidence-Based Review of the Use of the Pulmonary Artery Catheter: Impact Data and Complications. Crit Care 2006, 10 Suppl 3, doi:10.1186/CC4834.
- Thorson, C.M.; Van Haren, R.M.; Ryan, M.L.; Pereira, R.; Olloqui, J.; Guarch, G.A.; Barrera, J.M.; Busko, A.M.; Livingstone, A.S.; Proctor, K.G. Admission Hematocrit and Transfusion Requirements after Trauma. J Am Coll Surg 2013, 216, 65–73, doi:10.1016/J.JAMCOLLSURG.2012.09.011.
- Nijboer, J.M.M.; Van Der Horst, I.C.C.; Hendriks, H.G.D.; Ten Duis, H.J.; Nijsten, M.W.N. Myth or Reality: Hematocrit and Hemoglobin Differ in Trauma. J Trauma 2007, 62, 1310–1312, doi:10.1097/TA.0B013E3180341F54.
- Tomizawa, M.; Shinozaki, F.; Hasegawa, R.; Shirai, Y.; Motoyoshi, Y.; Sugiyama, T.; Yamamoto, S.; Ishige, N. Low Hemoglobin Levels Are Associated with Upper Gastrointestinal Bleeding. Biomed Rep 2016, 5, 349, doi:10.3892/BR.2016.727.
- Bruns, B.; Lindsey, M.; Rowe, K.; Brown, S.; Minei, J.P.; Gentilello, L.M.; Shafi, S. Hemoglobin Drops within Minutes of Injuries and Predicts Need for an Intervention to Stop Hemorrhage. J Trauma 2007, 63, 312–315, doi:10.1097/TA.0B013E31812389D6.
- Jubran, A. Pulse Oximetry. Crit Care 2015, 19, doi:10.1186/S13054-015-0984-8.
- Jarabin, J.; Bere, Z.; Hartmann, P.; Tóth, F.; Kiss, J.G.; Rovó, L. Laser-Doppler Microvascular Measurements in the Peri-Implant Areas of Different Osseointegrated Bone Conductor Implant Systems. Eur Arch Otorhinolaryngol 2015, 272, 3655–3662, doi:10.1007/S00405-014-3429-0.
- Greksa, F.; Butt, E.; Csonka, E.; Jávor, P.; Tuboly, E.; Török, L.; Szabo, A.; Varga, E.; Hartmann, P. Periosteal and Endosteal Microcirculatory Injury Following Excessive Osteosynthesis. Injury 2021, 52 Suppl 1, S3–S6, doi:10.1016/J.INJURY.2020.11.053.
- Jávor, P.; Rárosi, F.; Horváth, T.; Török, L.; Varga, E.; Hartmann, P. Detection of Exhaled Methane Levels for Monitoring Trauma-Related Haemorrhage Following Blunt Trauma: Study Protocol for a Prospective Observational Study. BMJ Open 2022, 12, doi:10.1136/BMJOPEN-2021-057872.
- Mészáros, A.T.; Szilágyi, ágnes L.; Juhász, L.; Tuboly, E.; érces, D.; Varga, G.; Hartmann, P. Mitochondria As Sources and Targets of Methane. Front Med (Lausanne) 2017, 4, doi:10.3389/FMED.2017.00195.
- Tuboly, E.; Molnár, R.; Tokés, T.; Turányi, R.N.; Hartmann, P.; Mészáros, A.T.; Strifler, G.; Földesi, I.; Siska, A.; Szabó, A.; et al. Excessive Alcohol Consumption Induces Methane Production in Humans and Rats. Scientific Reports 2017 7:1 2017, 7, 1–10, doi:10.1038/s41598-017-07637-3.
We highly agree that real numbers would be preferred over a scale from 1 to 3. However, reviewing the literature we observed significant differences between different studies from different countries; therefore, we favored the careful approach of highlighting trends instead of exact numbers.
Ranking the markers and methods according to the average time to availability is a judicious idea, and we would suggest the following ranking (from shortest to longest time): HR, SpO2, SBP < eFAST < Hb, Hct, BD, lactate < urinary output < CT angiography < pulmonary artery pressure < sublingual mucosal microcirculation < mesenterial blood flow, mesenteric mucosal microcirculation. It is important to mention, that sublingual videomicroscopy is a rapidly performable technique; however, evaluation and quantification takes time. Regarding mesenterial blood flow (slightly simplified and practically speaking: the superior mesenteric artery flow), time to availability depends on whether we accept monitoring exhaled CH4 concentrations as a valid technique for following the alterations of the perfusion of the superior mesenteric artery. Currently, clinical studies are in progress in our institution to clarify this.
We agree that cost-benefit is a crucial aspect in medicine. Quantifying the cost-benefit ratio of the diagnostic markers and methods described in our current study would be hardly performable; nevertheless, we believe that the following methods hold a cost-benefit proportion greater than 1.0: HR, SpO2, SBP, eFAST, Hb, Hct, BD, lactate, urinary output, CT angiography, pulmonary artery pressure. In this issue too, if we accept monitoring exhaled CH4 concentrations as a valid technique, we believe that “mesenterial blood flow” also belongs to this list.
Although not all trauma centers are “created equal”, we believe that the following are available in any trauma center: HR, SpO2, SBP, eFAST, Hb, Hct, BD, lactate, urinary output, CT angiography, pulmonary artery pressure.
With regards to the learning curve of the users, we think it is reasonable to talk only about the techniques that require skill to perform. Again, techniques that are currently in experimental phase (mesenterial blood flow, mesenteric mucosal microcirculation) would be difficult to insert into a list based on the learning curve of users. As clinicians, we highly value eFAST for the short and steep learning curve, although it is to emphasize that just like any examination withing the sonography modality, the accuracy of eFAST is also highly dependent on the patient and the user. We believe that evaluations of CT angiography and microcirculatory recordings entail a somewhat less steep learning curve. We also believe that the obtainment and evaluation of pulmonary artery pressure values require the most skill and practice.
Round 2
Reviewer 2 Report
The authors have mostly addressed my concerns. However, I think the authors should report the full number of manuscripts reviewed, which seems to be far more than 34.
Author Response
Re: Ms. jcm-2386209 - " Impairment of mesenteric perfusion as a marker of major bleeding intrauma patients " by P. Jávor et al.
Reviewer 2
1.) The authors have mostly addressed my concerns. However, I think the authors should report the full number of manuscripts reviewed, which seems to be far more than 34.
Thank you for this remark. Indeed, altogether many more manuscripts were reviewed. We mustered all utilized papers and classified them based on the way of their access. As a result, we can state that 57 manuscripts were reached via the reference lists of the 34 papers accessed through our MEDLINE search. Forty-nine studies related to the topic were already known by the authors. Nine articles were suggested by other experts the authors briefly consulted with. Twelve studies were accessed through non-systematic use of online search engines. In total, the authors reviewed 161 full-text articles.
Our manuscript has been extended accordingly (page 5, line 212-218): “The information obtained from these studies was critically evaluated and extended with data extracted from further papers. Fifty-seven manuscripts were reached via the reference lists of the 34 papers accessed through the above-mentioned MEDLINE search. Forty-nine studies related to the topic were already known by the authors. Nine articles were suggested by other experts the authors briefly consulted with. Twelve studies were accessed through non-systematic use of online search engines. In total, the authors reviewed 161 full-text articles.”